# Iatrogenic Barotrauma in COVID-19-Positive Patients: Is It Related to the Pneumonia Severity? Prevalence and Trends of This Complication Over Time

**DOI:** 10.3390/biomedicines10102493

**Published:** 2022-10-06

**Authors:** Nicola Maggialetti, Stefano Piemonte, Emanuela Sperti, Francesco Inchingolo, Sabrina Greco, Nicola Maria Lucarelli, Pierluigi De Chirico, Stefano Lofino, Federica Coppola, Claudia Catacchio, Anna Maria Gravili, Angela Sardaro, Amato Antonio Stabile Ianora

**Affiliations:** 1Department of Medical Science, Neuroscience and Sensory Organs (DSMBNOS), University of Bari “Aldo Moro”, 70124 Bari, Italy; 2Interdisciplinary Department of Medicine, Section of Radiology and Radiation Oncology, University of Bari “Aldo Moro”, 70124 Bari, Italy; 3Interdisciplinary Department of Medicine, Section of Dental Medicine, University of Bari “Aldo Moro”, 70124 Bari, Italy

**Keywords:** barotrauma, pneumomediastinum, pneumothorax, mechanical ventilation, pneumonia severity score, COVID-19

## Abstract

COVID-19 has attracted worldwide attention ever since the first case was identified in Wuhan (China) in December 2019 and was classified, at a later time, as a public health emergency of international concern in January 2020 and as a pandemic in March 2020. The interstitial pneumonia caused by COVID-19 often requires mechanical ventilation, which can lead to pulmonary barotrauma. We assessed the relationship between pneumonia severity and the development of barotrauma in COVID-19-positive patients mechanically ventilated in an intensive care unit; we therefore analyzed the prevalence of iatrogenic barotrauma and its trends over time during the pandemic in COVID-19-positive patients undergoing mechanical ventilation compared to COVID-19-negative patients, making a distinction between different types of ventilation (invasive mechanical ventilation vs. noninvasive mechanical ventilation). We compared CT findings of pneumomediastinum and pneumothorax in 104 COVID-19-positive patients hospitalized in an intensive care unit and 101 COVID-19-negative patients undergoing mechanical ventilation in the period between October 2020 and December 2021. The severity of pneumonia was not directly correlated with the development of barotrauma. Furthermore, a higher prevalence of complications due to barotrauma was observed in the group of mechanically ventilated COVID-19-postive patients vs. COVID-19-negative patients. A higher rate of barotrauma was observed in subgroups of COVID-19-positive patients undergoing mechanical ventilation compared to those treated with invasive mechanical ventilation. The prevalence of barotrauma in COVID 19-positive patients showed a decreasing trend over the period under review. CT remains an essential tool in the early detection, diagnosis, and monitoring of the clinical course of SARS-CoV2 pneumonia; in evaluating the disease severity; and in the assessment of iatrogenic complications such as barotrauma pathology.

## 1. Introduction

COVID-19 has attracted worldwide attention ever since the first case was identified in Wuhan (China) in December 2019 and was classified, at a later time, as a public health emergency of international concern in January 2020 and as a pandemic in March 2020; it quickly spread globally, causing interstitial pneumonia that often required mechanical ventilation. The diagnosis of COVID-19 is typically performed on biological samples through a PCR test for COVID-19 detection. However, the role of radiology is essential in the management of this pathology. HRCT is indeed the most accurate technique in identifying most typical findings of COVID-19 disease: focal or multifocal ground-glass opacities (GGOs), consolidation, nodules, reticulation, thickening of interlobular septa, crazy paving appearance, traction bronchiectasis, bronchovascular thickening in the lesion, air space consolidation, and subpleural lines [1].

CT also allows us to assess the severity of the infection, including its atypical, unexpected manifestations, such as lymphadenopathy, pleural effusion or pericardial effusion, and multi-organ involvement. It is also essential to monitor disease progression. In certain cases, it can confirm diagnosis (sensitivity ≃ 98%) in suspected patients with false-negative laboratory results [2]. There have also been some predictive models showing the temporal progression of COVID-19 disease through chest CT findings that have been identified: in the early stage, COVID-19 lesions are relatively localized and limited to the subpleural or peribronchovascular regions, showing patchy or segmental pure GGOs (ground-glass opacities). In the progressive phase, CT mainly shows an increase in GGOs’ involvement in lobes and consolidations, with thickening of interlobular septa and a crazy paving appearance as common CT findings. In the advanced stage, CT manifestations mainly include diffuse bilateral parenchymal consolidation surrounded by ground-glass opacification with pulmonary parenchymal bands and, sporadically, pleural effusion [3].

One of the most frequent and significant complications of COVID-19 is ARDS (acute respiratory distress syndrome). ARDS is generally defined by the acute onset of hypoxaemia and bilateral pulmonary infiltrates: it is classified according to the PaO_2_ to FiO_2_ ratio (ratio expressed as X in the following brackets) and can be mild (200 mmHg ≤ X ≤ 300 mmHg), moderate (100 mmHg ≤ X ≤ 200 mmHg), and severe (X < 100 mmHg) [4,5].

In patients with COVID-19 and acute hypoxiemic respiratory failure, conventional oxygen therapy might be insufficient, and a non-physiological and invasive treatment that can be life-saving may be required in severe ARDS. In order to provide enhanced respiratory support, available therapies range from high-flow nasal cannula (HFNC) oxygen to mechanical ventilation, which can be administered through invasive techniques (IMV) such endotracheal tube (ETT) or tracheostomy tube (TT) or non-invasive techniques (NIV) such as positive airway pressure through face or nasal mask (CPAP, continuous positive airway pressure). According to COVID-19 treatment guidelines, for adults with COVID-19 and acute hypoxemic respiratory failure despite conventional oxygen therapy, the first line of treatment is HFNC oxygen; if patients fail to respond, NIV or intubation should be initiated [6]. Mechanical ventilation is recommended in patients presenting moderate to severe ARDS. A higher positive end-expiratory pressure (PEEP) is suggested, and in the case of refractory hypoxemia despite optimized ventilation, prone ventilation lasting 12 to 16 h per day [7]. Prone positioning could prevent lung injury caused by ventilators because it reduces ventral alveolar expansion and dorsal alveolar collapse, resulting in more homogeneous ventilation. The difference between dorsal and ventral transpulmonary pressure is reduced, allowing minor lung compression and enhanced perfusion [8]. ARDS and patients who are severely hypoxemic (Pao2: Fio2 ratio < 150 mm Hg, Fio2 ≥ 0.6, PEEP ≥5 cmH_2_O) can benefit from prone positioning if early intervention is performed, and positioning is maintained for relatively long sessions [9]. However, an unnecessarily high PEEP can increase the risk of lung overdistension.

According to pathophysiology, it has been shown in other studies that there is a proportional association between cytokine production, PEEP, and tidal volume [10]. These results suggest that higher PEEP and higher tidal volume may further increase the cytokine response with the worsening of alveolar damage, predisposing patients to barotrauma. Therefore, mechanical ventilation can the increase risk of developing pulmonary barotrauma (pneumomediastinum, pneumothorax, and subcutaneous emphysema), which occurs due to alveolar rupture caused by high transalveolar pressure. The last intervention line for the treatment of refractory respiratory failure and severe ARDS is represented by extracorporeal membrane oxygenation (ECMO). It is an invasive technique performed in tertiary care centers that allows the oxygenation of the blood while removing CO_2_ at the same time. This approach gives time for the failing lung to recover [11,12]. ECMO guidelines for COVID-19-related ARDS were based on pre-COVID-19 trials, and ECMO was started in patients <71 years old with severe initial presentation and a short duration of mechanical ventilation (MV) before ECMO (i.e., <7 or <11 days) [13,14]. Data on ECMO efficacy in COVID-19 related ARDS are limited and mainly come from case reports or from the experiences of single centers. The reported mortality rate was 39% (95% CI 34–43) [15,16].

Pneumomediastinum refers to a gaseous infiltration of the mid-thoracic cellular tissues consequent to the penetration of air into the mediastinal space. This occurs according to “Hamman–Macklin mechanism”, where “primum movens” are represented by the rupture of the alveoli in contact with the pulmonary vessels, interstitial tissue, small bronchi, and bronchioles, where pressure is higher. Air penetrates the interstitium, and from here, it runs through the perivasal sheaths and the peri-bronchial fascial planes and reaches the pulmonary hilum, resulting in pneumomediastinum. From the hilum, the air can distribute itself cranially along the vascular sheaths of the neck (most frequent occurrence), causing the formation of subcutaneous emphysema of the supraclavicular, axillary, cervical, face, and thoracic regions. In other, less frequent cases, the air can distribute itself caudally through the diaphragmatic orifices, causing pneumoperitoneum and retropneumoperitoneum. Sometimes, it can peel off the parietal pleura, causing pneumothorax, which refers to air collection in the pleural cavity between the visceral and parietal pleura: in the case of mediastinal parietal pleura caused by gaseous infiltration due to pneumomediastinum, pneumothorax appears as a consequence of pneumomediastinum [17,18].

A relevant incidence of barotrauma in hospitalized COVID-19-positive patients has been reported in several studies [19,20]. In COVID-19-positive patients, the pulmonary air space may be more vulnerable to alveolar damage because of the significant increase in trans-alveolar pressure over the local stress–strain threshold that guarantees epithelial and interstitial integrity; additionally, clinical manifestations such as cough and the increased energy expended to inhale and exhale (work of breathing) increase the stress applied to the respiratory system, which may even aggravate lung damage by several mechanisms gathered under the name “patient self-inflicted lung injury”. Lung inflammation associated with the derecruitment of some alveolar segments, which occurs in moderate to severe COVID-19 disease, and variation in negative pleural pressure caused by improved work of breathing induce locally concentrated stress amid ventilated and collapsed alveolar segments. Better expanded lung units recruit partially open alveoli with supra-physiological transalveolar pressure, which can lead to heterogeneous deformation stress over the alveolar membrane [21,22].

According to these studies, interstitial pneumonia caused by COVID-19 predisposes mechanically ventilated patients to barotrauma: our aim was to assess the relationship between pneumonia severity and the development of barotrauma in COVID-19-positive patients who had been mechanically ventilated in an intensive care unit (ICU). Moreover, we evaluated the prevalence of iatrogenic barotrauma in COVID-19-positive patients compared to COVID-19-negative patients undergoing mechanical ventilation: we have also classified COVID-19-positive patients with barotrauma according to types of ventilation (IMV vs. NIV) and analyzed the average time of onset of this complication and its trends over time during the period of study.

## 2. Materials and Methods

The study was conducted by the University Hospital “Policlinico of Bari” and the “Department for maxi emergencies—Fiera del Levante”, Bari, Italy.

### 2.1. Study Sample

The research involved 205 patients admitted to the ICU from October 2020 to December 2021. Patients were divided into two groups (Figure 1): COVID-19-positive patients (*n* = 104, male: 76, female: 28, mean age: 65) and COVID-19-negative patients (*n* = 101, male: 64, female: 37, mean age: 59). COVID-19-positive patients were again divided into two subgroups according to the type of mechanical ventilation: ETT (n. 94) and CPAP-mask (*n*. 10), in order to observe differences in barotrauma incidence.

Inclusion criteria were (a) patients admitted to ICU undergoing mechanical ventilation; (b) included patients were subsequently divided into two groups “COVID+” and “COVID−” according to the result of a PCR test for SARS-CoV-2 infection from biological samples; (c) among COVID+ patients, we selected patients with a pneumonia severity score >10 based on a semi-quantitative CT score system.

Exclusion criteria were (a) patients diagnosed with pre-existing barotrauma at admission; (b) patients who were not on mechanical ventilation; and (c) patients with a pneumonia severity score <10 (Figure 2).

Several algorithms were developed by considering the severity of the pulmonary involvement of COVID-19 according to CT imaging in order to properly standardize and quantify the radiological examinations. The “chest CT score” is a method developed by Li et al. [12]. In this study, the pneumonia CT severity score was assessed, awarding each lobe a score (0 to 5) related to lobar involvement expressed as a percentage value: a score 0 corresponds to an absence of involvement, a score 1 shows involvement of less than 5%, a score of 2 shows involvement from 5% to 25%, a score of 3 shows involvement from 26% to 49%, a score of 4 shows involvement from 50% to 75%, and a score 5 shows involvement greater than 76%. Calculated for each lobe of both lungs, the sum of each score reaches a maximum of 25 points [19].

Diagnosis of pneumomediastinum/pneumothorax was confirmed by computed tomography (CT), while the average time of onset of barotrauma was estimated from the beginning of mechanical ventilation in the ICU.

### 2.2. Scanning Protocol

Siemens Somatom Definition DS CT scanner (Erlangen, Germany) was used to perform CT exams. Scanning protocols followed the same acquisition parameters: slice thickness 0.75 mm, tube voltage 100 kVp, 38 mAs, rotation time 0.33 s, pitch 1.1. All images obtained were elaborated with reconstruction of 1 mm slice thickness.

### 2.3. Imaging Assessment

All CT images obtained were archived through the institutional PACS (Carestream Health, Rochester, NY, USA). MPR (multiplanar reformatting) and 3D MIP (3D maximum intensity projection) were performed in order to analyze images using both mediastinal and lung windows. Visual analysis of CT images was conducted independently by two radiologists (A.A.S.I. and N.M., with 23 and 12 years of experience, respectively), and any disagreement was discussed and solved with the consensus of both radiologists.

### 2.4. Statistical Analysis

All statistical analyses were conducted using JMP statistical software.

We used the T-test to compare the averages of continuous variables such as age and to compare the average time of onset of barotrauma in the COVID+ patients, differentiating the group with PNM and the group with PNX.

The comparison between the average values of the severity score in the group with barotrauma compared to that with no barotrauma was also analyzed through a T-test.

We used the Chi-square test to compare the prevalence of barotrauma in the group of COVID+ patients vs. COVID− patients.

A contingency table was elaborated to compare the incidence of barotrauma in the NIV vs. MIV group: a Pearson Chi-square test was then performed.

We used weighted least-squares regression to assess the number of cases of barotrauma over time: an ANOVA was conducted on the data.

## 3. Results

### 3.1. Prevalence of Barotrauma

#### 3.1.1. Prevalence of Barotrauma in COVID+ vs. COVID−

Prevalence of barotrauma was evaluated between the COVID+ group vs. the COVID− group during the entire period of study (Figure 3): a higher prevalence of complications due to barotrauma was observed in the group of mechanically ventilated COVID+ patients, with an incidence of pneumomediastinum of 15% (16/104) and of isolated pneumothorax of 14% (15/104). In COVID− patients, there was an incidence of pneumomediastinum of 3% (3/101) and of isolated pneumothorax of 8% (8/101). A Chi-square test was performed to compare percentages between the two groups (*p*: 0.02). We found cases of pneumomediastinum complicated with pneumothorax exclusively in the COVID+ group (Figure 4).

#### 3.1.2. Relation between the Prevalence of Barotrauma in COVID+ Group and Type of Mechanical Ventilation

In the COVID+ group, the presence of barotrauma was classified according to the type of mechanical ventilation in use: NIV through C-PAP mask caused a higher rate of barotrauma (60% vs. 27% in the subgroup IMV with ETT). A contingency table was made to compare the results: a Pearson Chi-square test was performed (*p*: 0.02, Figure 5).

### 3.2. Relation between Barotrauma and Pneumonia Severity Score in COVID+

In the COVID+ group, the mean value of the pneumonia severity score was compared in the subgroup “with barotrauma” vs. “no barotrauma”. We did not find any significant difference in the pneumonia severity scores in COVID+ with barotrauma vs. no barotrauma (COVID+ pneumonia severity mean score: 21 vs. 20 in COVID−, Figure 6). A T-test was performed to compare the average values among the two groups (*p*: 0.01).

### 3.3. Average Time of Onset of Pneumothorax and Pneumomediastinum

In the COVID+ group, pneumomediastinum developed on average within 3 days, earlier than pneumothorax, for which the average time was 18 days (Figure 7). A *t*-test was performed to compare average values among the two groups (*p*: 0.0009).

### 3.4. Trend of Barotrauma in COVID+ Group vs. COVD− Group

Cases of barotrauma were monitored during the period of observation. We observed a descending trend in the barotrauma rate in COVID+ patients. ANOVA was performed (*p*: 0.017). The incidence of barotrauma in COVID− patients did not differ (Figure 8).

## 4. Discussion

In most cases, SARS-CoV2 disease leads to severe interstitial pneumonia, which modifies the normal lung architecture and decreases compliance; in addition, the necessity to mechanically ventilate patients with high flows and pressures exposes them to a higher risk of barotraumatic pathology (Figure 9) compared to patients who are mechanically ventilated for other diseases, probably due to the lower compliance of the affected lungs.

This is exactly what we observed: we found a higher prevalence of complications due to barotrauma in the group of mechanically ventilated COVID+ patients. This alteration seems to correlate not only with an increased probability of developing both pneumomediastinum and pneumothorax as manifestations of barotraumatic pathology, but also with an increased risk of developing complicated conditions (pneumomediastinum associated with pneumothorax), an event that was not observed in patients without SARS-CoV2 pneumonia in our study.

Nevertheless, we did not find any significant differences in the pneumonia severity scores in COVID+ patients with barotrauma vs. no barotrauma, suggesting that the severity of pneumonia is not a factor that directly predisposes patients to barotrauma.

According to our results, NIV caused a higher rate of barotrauma. This could be due to several mechanisms. Patients under NIV did not receive ventilation in prone position; therefore, a prone position, often performed on IMV patients with the aim to better ventilate the posterior segments of the lungs, could have a protective effect in reducing barotrauma. Another protective factor in the patients who underwent IMV could be that they need to receive specific therapies with the aim to reduce the resistance of the lung and chest wall. Consequent lower lung resistances could reduce the barotrauma rate by reducing alveolar stress–strain. Finally, people under NIV were sustaining spontaneous breathing: dyspnea or increased work of breathing, due to hypoxia increasing the maximal inspiratory effort, leading to an increase in transpulmonary pressure, especially when under mechanical ventilation with continuous positive airway pressure (C-PAP). Therefore, in these conditions, patients’ lungs under NIV could have been more exposed to the high pressures of mechanical ventilation, a factor promoting barotrauma.

As mentioned above in the introduction, pneumomediastinum can be seen as the first manifestation of iatrogenic barotrauma and is caused by alveolar rupture and gaseous infiltration from the peribronchial sheaths up to the mediastinal planes. This is what emerged from our study: pneumomediastinum developed on average after 3 days of mechanical ventilation. Most cases of pneumomediastinum that we observed were associated with pneumothorax (10/16), with pneumothorax as an evolution of barotrauma pathology that started with pneumomediastinum. However, when pneumothorax was isolated, its onset occurred after 18 days on average, suggesting that it was promoted by other factors, such as the evolution of pneumonia.

The descending trend in barotrauma incidence we observed could be explained by a better knowledge of COVID-19′s physiopathology and better experience in the management of these patients, especially regarding ventilation techniques; mechanical ventilation may not be standardized, but it should be adapted to single cases while being aware of the possibility of barotrauma complications. In the COVID− patients, we did not find a significant change in the incidence during the entire study period.

## 5. Conclusions

Based on our experience, the incidence of barotrauma in patients who were mechanically ventilated was significantly higher in COVID+ patients than in COVID− patients, regardless of the degree of the severity of pneumonia. Therefore, it would be useful to identify other risk factors that expose mechanically ventilated patients to iatrogenic barotrauma. The inflammatory response causing acute lung injury in COVID-19 patients remains a challenge for invasive and lung-protective ventilation management. CT remains an essential tool in the early detection, diagnosis, and monitoring of the clinical course of SARS-CoV2 pneumonia in evaluating the disease severity, but also in the assessment of iatrogenic complications such as barotrauma pathology.

## Figures and Tables

**Figure 1 biomedicines-10-02493-f001:**
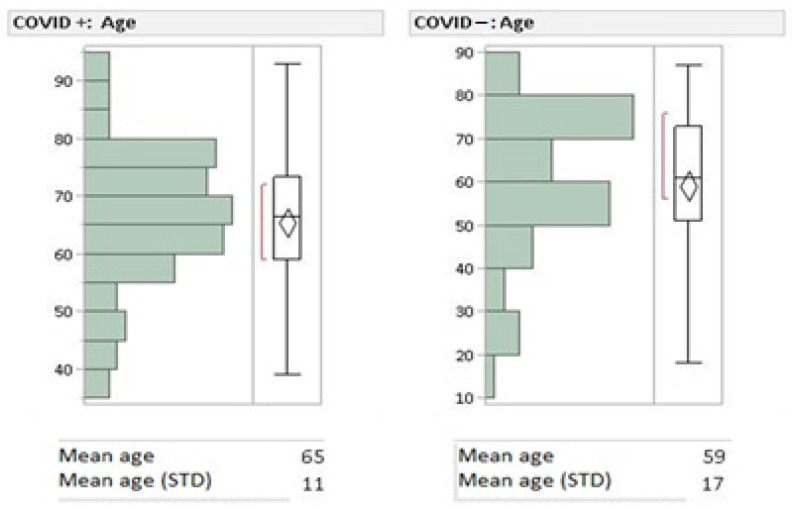
Population mean age and standard deviation (STD).

**Figure 2 biomedicines-10-02493-f002:**
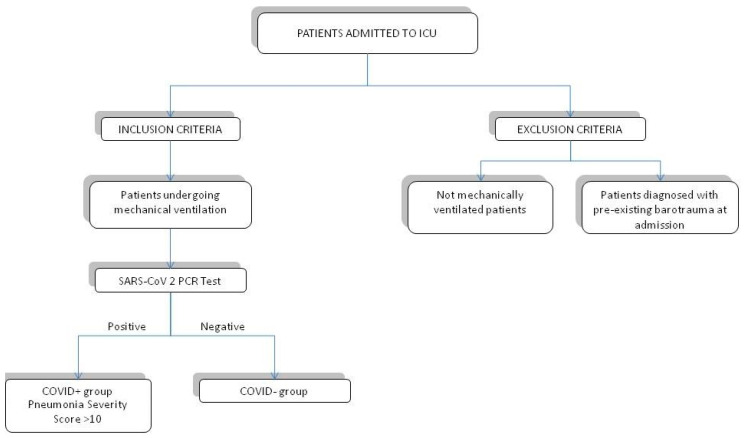
Flowchart: inclusion and exclusion criteria.

**Figure 3 biomedicines-10-02493-f003:**
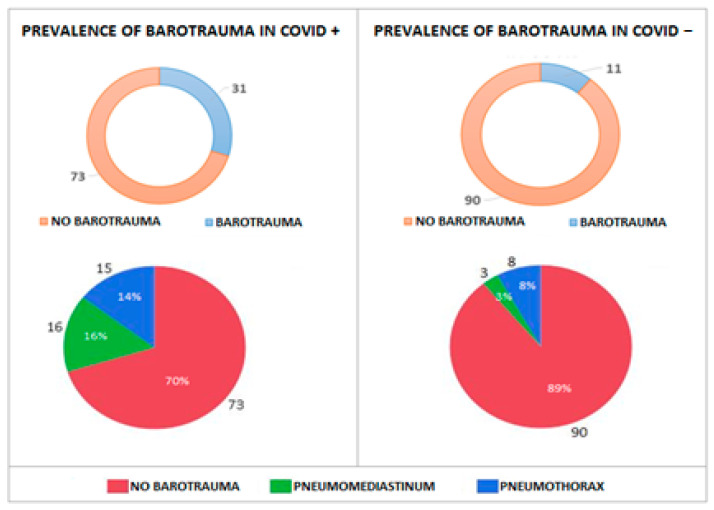
Prevalence of barotrauma in COVID+ patients vs. COVID− patients in ICU.

**Figure 4 biomedicines-10-02493-f004:**
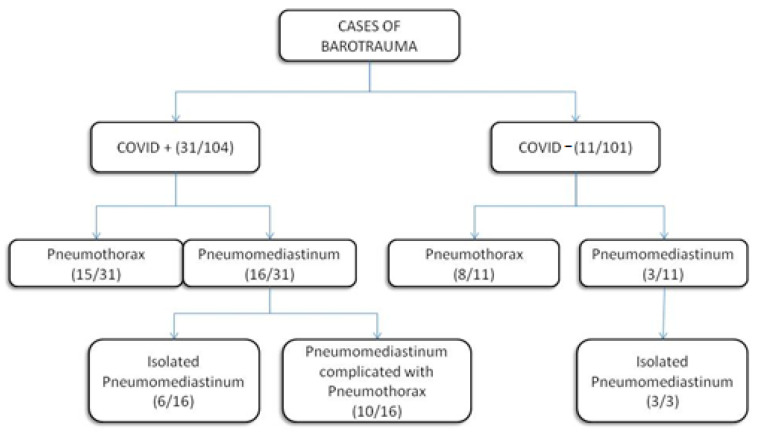
Types of barotrauma among groups.

**Figure 5 biomedicines-10-02493-f005:**
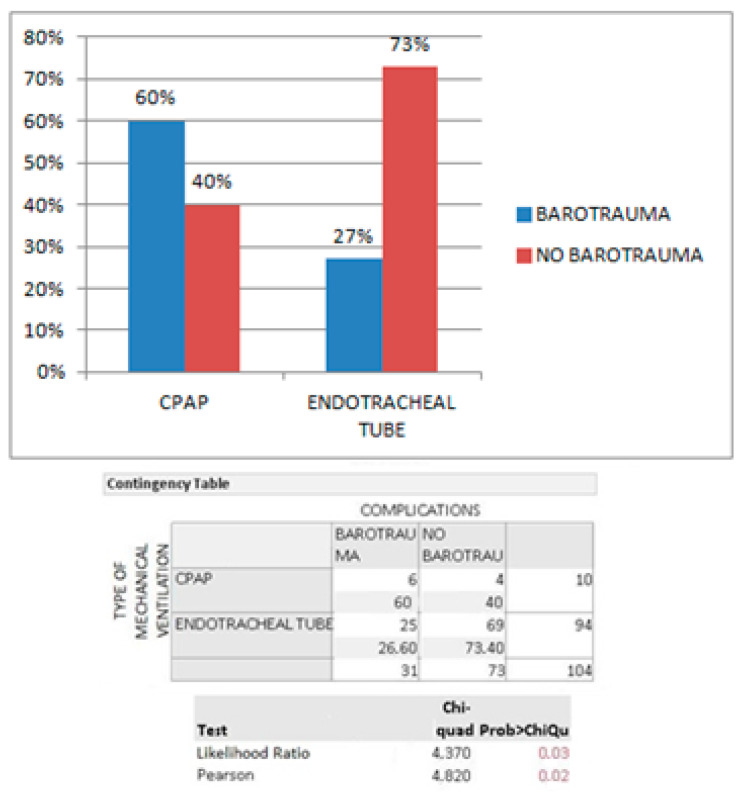
Presence of barotrauma according to type of ventilation in COVID+ patients: *p*-value is reported in red.

**Figure 6 biomedicines-10-02493-f006:**
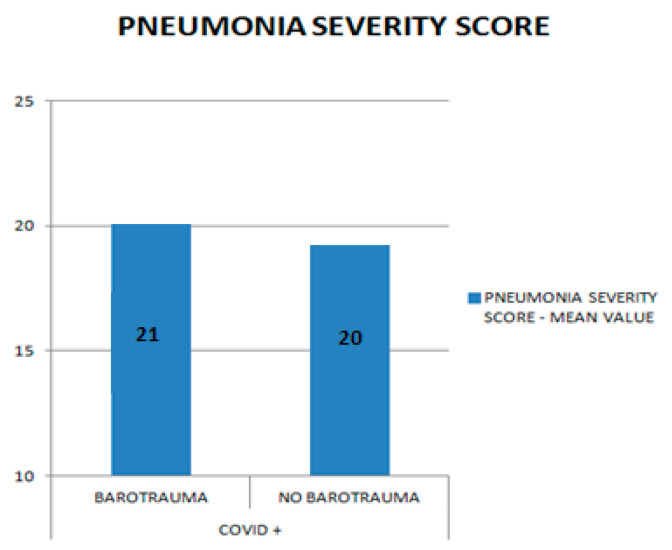
Comparison between barotrauma and pneumonia severity score in COVID+ patients.

**Figure 7 biomedicines-10-02493-f007:**
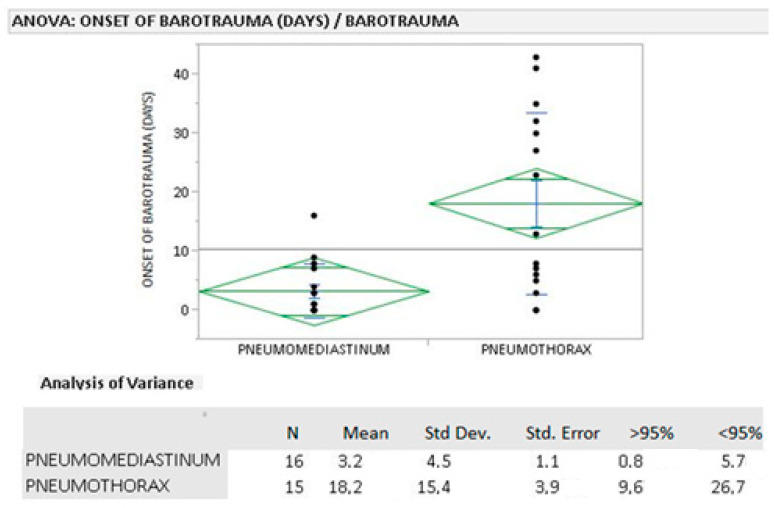
Average time of onset of barotrauma in COVID+ patients.

**Figure 8 biomedicines-10-02493-f008:**
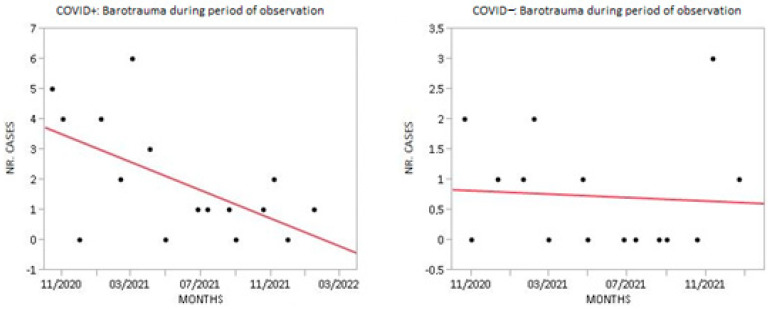
Cases of barotrauma in COVID+ group vs. COVD- group during period of observation.

**Figure 9 biomedicines-10-02493-f009:**
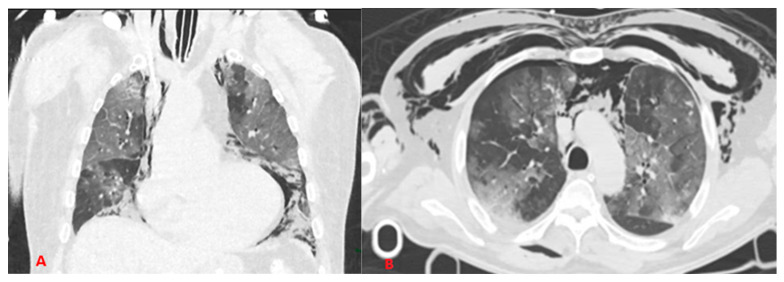
Examples of barotrauma: (**A**) coronal scans show pneumomediastinum associated with subcutaneous emphysema in COVID+ patient with ETT. (**B**) Axial scans show severe pneumomediastinum associated with pneumothorax and subcutaneous emphysema in COVID+ patient with CPAP.

## Data Availability

Data is contained within the article.

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
