# Peer review of "Iatrogenic Barotrauma in COVID-19-Positive Patients: Is It Related to the Pneumonia Severity? Prevalence and Trends of This Complication Over Time"

_biomedicines, 2022, doi:10.3390/biomedicines10102493_

Round 1

Reviewer 1 Report

Major question

1. Is there any information about existence of COPD findings with CT scan as the the background lung diseases and its relationship with barotrauma?

2. COVID19 shows higher risk of barotrauma than others and this is coincide with other articles but statistical analysis is necessary, especially multi variate analysis to show the significance of COVID19 diseases. Also the reasoning of the difference between intubation and NIV needs statistical analysis.

Minor question

Figure 4 and 5 shows overlapping information. 

Author Response

Dear Reviewer 1,

Thank you very much for your previous comments that helped us improve this manuscript.

Your question about the existence of COPD findings with CT scan and its relationship with barotrauma is really interesting. However, we have no data concerning the presence of COPD because patients involved in our study were all from ICU, therefore their clinical informations related to an eventual pre-existing COPD  were not always taken into consideration in reason of their critical condition. Moreover CT typical patterns of COPD were often hidden by several lung injuries due to COVID-19.

According to your suggestions about the need of statistical analysis, we produced a new paragraph (“Statistical Analysis”) in the paper relating to it; in this paragraph the analysis conducted are explained in details.

We hope that you will find our manuscript acceptable in its present form.

Reviewer 2 Report

The study is devoted to a serious problem. Indeed, when accompanying patients with COVID-19, barotrauma occurs more often.

It is clear that the clinical trial is limited by the accepted protocols of patient management. At the same time, a critical remark is the different sample sizes (n=10 and n=94) for different types of mechanical ventilation. The differences are not statistically processed.

Research on such clinical material is of great interest.

It is necessary to conduct a more detailed analysis and correlate the probability of developing barotrauma with 1. The development of a cytokine storm, 2. The parameters of mechanical ventilation (https://www.ncbi.nlm.nih.gov/pmc/articles/PMC7336751 / Barotrauma risk is particularly important to recognize as these critically ill patients may be managed by staff less familiar with the management of ventilator settings.)

A critical area of analysis is to identify the causes of a decrease in the number of barotrauma over time. In fact, we are talking about the fact that new guidelines on mechanical ventilation of the lungs should be presented to prevent similar problems in the future. So the question in the study is not worth it, although the trend of reducing barotrauma has been found. What are its causes?

The article should be reworked, the drawings rethought, statistical processing of the material carried out. And most importantly, modifications of medical procedures to improve the management of such patients were discussed. In itself, the fact of a greater number of barotrauma in COVID-19 has been documented earlier.

Author Response

Dear Reviewer 2,

Thank you very much for reviewing our manuscript. Your insights led to improvement of our paper.

As you suggested, we produced a new paragraph in the paper (“Statistical Analysis”) that satisfies the need of statistical analysis; in this paragraph the analysis conducted are explained in details. Specifically, in regards to the impact of different types of mechanical ventilation on barotrauma prevalence, a contingency table was made to compare the incidence of barotrauma in the NIV vs MIV group, subsequently it was applied the Pearson Chi-square test.

Based on the results we obtained, there is great interest in investigating the impact of ventilation modes as well as the influence of the cytokine storm on the development of barotrauma: a dedicated study, primarily based on clinical aspects, that involves a numerically more significant sample should be performed. In our paper we focused on the more purely radiological aspects, analyzing the pulmonary patterns of COVID-19 and evaluating their influence in the eventual development of barotrauma.

Regarding the causes of a decrease in the number of barotrauma over time, we reported our considerations in the paragraph “Discussion” of the paper. However we cannot identify the certain cause of this trend. We agree in the importance of defining new guidelines aimed at preventing barotraumatic pathology during mechanical ventilation, for this reason a more in-depth and targeted study should be conducted, which takes into consideration, for example, ventilation modes, ventilation parameters, hospital wards and experience of the operators.

In conclusion, we have modified the introduction, the drawings and carried out a statistical processing of the results. We hope that you will find our paper suitable for publication in its current form.

Round 2

Reviewer 1 Report

Statistical analysis is well revised.

Author Response

Dear Reviewer 1,

we are really glad that you appreciate this latest version of our manuscript. Your insights led to improvement of our paper.

Reviewer 2 Report

A revised version of the article may be published. Corrections and explanations are sufficient.

Author Response

Dear Reviewer 2,

we are glad that you appreciate this last version of our manuscript. We really thank you for your observations that helped to improve quality of our work.